# A Collector Deceives—About the Ways of Deceiving Women by Men and Men by Women as far as Spending Money on Collecting Items Is Concerned

**DOI:** 10.3390/ijerph192416755

**Published:** 2022-12-14

**Authors:** Tomasz Wirga, Anna Kopczak-Wirga

**Affiliations:** 1Department of Psychology, University of Opole, 45-040 Opole, Poland; 2Department of Sociology, University of Opole, 45-040 Opole, Poland

**Keywords:** collecting, deception, gender differences

## Abstract

The presented research shows that neither women nor men are honest with their partners when informing them about the amount of money spent on collecting items. Their behaviour may show signs of addiction to collecting. Men in comparison to women spend more and are less likely to lower the amounts of money spent on collected items. Those who earn more spend more on their collections. Women and men also use different techniques of hiding their expenses. Women do not inform about their expenses using denying techniques (such as saying that it was bought/borrowed a long time ago, etc.), whereas men inform about expenses but use preventive techniques (such as exchange). What is more, men tend to use a technique of lowering real costs “by a given amount”, whilst women use a technique “up to a given amount”; that is, they inform that they did not spend more than a given amount. In addition, the partners of collectors are aware that they are being cheated. On the other hand, collectors see the motivation for their lies in the misunderstanding of the hobby by their partners.

## 1. Introduction

It is possible to say that “collecting”, that is, spending money regularly on items, lies in human nature. Many of us collect creations of our civilisation such as stamps, coins, or postcards, creations of nature such as minerals or meteors, or simply want to have what both gives us pleasure and is important to us. Collectors are constantly looking for opportunities to increase their collections. An opportunity arises, for example, in a period of war. One of the many disadvantages of wars is stealing cultural property of the conquered country. It is possible to suspect that it has “always” been like that. However, when historical sources appeared, it allowed us to reach back to these actions which left a significant mark. As early as in the middle of the 12th century B.C., Shutruk-Nahhunte conquered and sacked Babylon. He forced the king Zababa-shuma-iddin to escape and as a result, he formed his own empire of Elam. This empire did not last long but what went down in history was the love for collecting. He took from Babylon, among others, the stela with the Code of Hammurabi from the 19th century, the stela of king of Akkad’s victory, Naramsima from the 22nd century B.C., many Hasidic boundary stones, the so-called kudurru from the 20th century B.C., or China vessels from tombs from the 15th century B.C. [1]. It is one of the first cases in history of collecting something which was not valuable, as it was not made of rare materials/metals, but had a certain value in the eyes of a collector. From this moment on, it is possible to follow numerous cases in history of chiefs, kings, or emperors who conquered and stole at the same time and collected works of art.

At the moment, there are stolen relics of ancient cultures in capital cities of many countries all over the world. In cities such as Paris, Madrid, Moscow, London, and Istanbul, there are visible remains of ambitions, actions, and collecting works of art. Collecting was a phenomenon not only at the state level—common citizens, the lives of whom were not saved in history, also collected items. Ciceron cited after Bias of Pirena the words “Omnia mea mecum porto,” which can be translated as “all that I need I carry on me.” Why is it that people sacrifice their resources, time, or social interactions to gain something which is unnecessary? Unfortunately, an answer to the question about the source of collecting is difficult and even impossible to find. According to Sontag [2], “A large private collection is a cluster of matter which is constantly jogging and causing the state of excessive liveliness. It is not only because it is always possible to add something to it, but because it, itself, is already an excess. The need of a collector is the need of an excess, luxury, wealth. A person who hesitates, who asks himself whether or not he needs something, is not a collector. A collection is always something more than it is needed”. This mechanism has existed and will always exist—this is aiming at possessing something for pleasure. It has existed since humankind appeared and will last until the end of human existence.

Another feature of collecting is that it is an element of respecting heritage and the past. It appears as early as in Ancient Rome [3] and reaches its apogee in the period of Enlightenment, under Luis XIV. Items have value themselves and are carriers of morality, ideology, and politics [4]. A contemporary aspect of collecting appears in its full grandeur in the year 1704 when the book entitled *Opticks* written by Newton inspired his contemporary theoreticians to observe, think, and imitate [5]. The fashion for having a microscope came to all European countries from France. Each scientist who wanted to be seen as an expert had to have it. The microscope became a symbol of knowledge, intelligence, or development. After some time, it became a value in itself, a desired object which was worth something only because one person had it and the other person did not. That is why people who wanted to have more than one microscope appeared in then-Europe. In 1736, a merchant from Paris called Edme-François Gersainta issued a catalogue of seashells and it started a fashion for collecting creations of nature [6]. A collector’s guide issued by Antoine Joseph Dezaillier d’Argenville in 1727 [7] became popular at that time. Everyone had to have it, even if he did not read it.

It should be noted that not everyone supported collecting. Critique appeared together with its popularisation. Rousseau was one of its fierce opponents. For him, collecting was thoughtless imitation, a consequence of bad taste and kind of like an addiction. It does not bring any social value; it is even harmful [8]: “it is easy to walk gathering sand and pebbles, fill pockets and your own collections with them, pretending to be a naturalist”. Collecting is a sign of vanity and artificial self-esteem because [8] “in order to study minerals in a useful way, it is necessary to be a chemist or a physicist, it is necessary to carry out experiments which are hard and cost a lot, it is necessary to work in a lab, spend a lot of money and time on being with carbon.” He also draws attention to a collector’s mind together with his feature—insatiable hunger for looking for new specimens which makes people addicted [8]: “I always begin with the most common plants that is with sandwort, chervil, borage and butterweed. I gather plants from the cage of my birds and with every new blade of grass which I find, I say to myself with satisfaction ‘it is one more plant’”. Collecting through its irresistible desire to possess causes an addiction that is difficult to overcome [9].

Despite criticism, the strand of collecting has constantly been increasing. When Napoleon went to Egypt, the fashion for everything coming from there came to Europe. According to Aristotle, curiosity was accompanied by the will to have more and more [10]—an excess of goods lies in human nature. Why? Because it gives us pleasure. The first fashion was the one for collecting items from Egypt, then from Greece, Rome, and so on. It is worth mentioning that according to Rousseau [10], the will to have another item is very strong. At the same time, the possibility of collecting is limited. That is why there is a conflict between resources and possibilities. It is necessary to remember that collecting, foraging, hobby, or other regular pleasures usually require set amounts of money because the need to possess leads the individual to depend on the collected objects and can be greater than rational thinking. It is a frequent practice that collectors, both women and men, do not tell the truth to their partners about the real amount of money they spend on their own hobbies.

What is a lie? It is interesting that in ancient times, a lie did not always have a negative meaning. Aristotle, in *The Nicomachean Ethics* [11], placed it in the middle, between the good and the bad. A little bit earlier, *Sophocles* in Philoctetes [12] writes that a lie does not have to be the object of contempt if it is able to save somebody. A change in its perception began thanks to Saint Augustin, for whom a lie [13] “is among others a statement made by somebody who wants to say something false in order to mislead, or wants something considered by him as false to be perceived by somebody else as something true.” Saint Thomas had a similar way of thinking. A lie is telling something else from what you think. For these two thinkers of the Church, a lie is evil because it is misleading. Nietzsche had a different opinion. For him, a lie is an indispensable element of life. If it did not exist, human life would not be possible. It allows us to create appearances which in turn allow us to bear the truth about life. Nietzche [14] wrote: “Falsity of an opinion is not for us an accusation against this opinion; people could not live without constant falsity—Renunciation of false beliefs would be renunciation of life.” There are many definitions of a lie in today’s psychology. It seems that the simplest definition, and at the same time, the most interesting, is provided by Masip [15]: “an intentional attempt of efficient, or not, hiding, faking and/or manipulating in any way information based on facts, or emotions, in order to create or keep in the receiver or receivers a belief about which the author knows that it is false.” What is interesting is the fact that it is not characteristic only of human species. In his book, Dawkins [16] describes many examples of behaviours of plants, insects, and mammals which can be interpreted as lies. It is possible to ask what the differences are among women and men in lying. The answer to this question is not unambiguous. Research on differences between men and women has a long and rich history. Gender differences in the scope of, e.g., mathematical skills [17], verbal skills [18], motor activity [19], personality differences [20], social behaviour [21], and many different spheres of functioning, including lies, were shown.

An interesting illustration (see Figure 1) of analysed dependencies is information found in social media. T-shirts with the following overprint “God when I die, do not allow my wife to sell my spinning wheels and fishing rods for the price I told her it had” are in regular sale.

This humorous picture shows the social perception of expenses on one’s own pleasures. However, do men spend more on their hobbies than women? In this place, it is possible to use another picture from social media (see Figure 2). This picture shows a garage which proposes the following: “Attention, as we know how hard it is to have both happiness in the garage and peace at home, we offer a new service for you! For extra PLN 9.99 (5 EUR), we will answer the phone when your wife is calling and we will confirm the amount of money which you told her that you had spent in our garage for parts for your vehicles!”

It certainly is a subjective approach. However, the sheer appearance of the memes on this subject can suggest that it is a quite common phenomenon. On 6 August 2017, the brand “Butik” wanted to make a joke and asked its clients whether it would be a good idea to introduce an option of a receipt for a husband which would decrease the value of the order by 50% (see Figure 3).

The idea itself brought a lot of comments. According to Długołęcka [22], a post like this would usually have 11 comments—this one had about 6 thousand of them. The comments were different. There were expressions such as “it is brilliant”; “such an option should be available in every shop”; “great idea”; as well as critical comments: “I do not lie to my man”; “I do not like this idea.” However, the most comments were like this: “The less he knows, the better sleep he gets, and it is necessary to care for the quality of sleep of our ‘second half’” or “There are expenses about which our man should not know, you will learn this after a few years of marriage.” There was a lot of interest concerning this idea and it was later present on different social media platforms (see Figure 4); thus, it shows that it is a trendy and interesting subject.

The aim of this research is to show how men and women (collectors/people who regularly spend money on their own pleasures) deceive their partners when informing them about the amount of money spent on their collections. What is more, we will try to describe techniques and ways of lowering the amounts spent on collecting items.

To sum up, we are going to answer the following research questions:

Do women and men hide their regular expenses on collecting items?

Are there gender differences in hiding one’s own regular expenses on collecting items?

Are there differences between collectors earning more and less than the national average in amounts regularly spent on collecting items?

Are there differences between men and women in techniques of hiding their regular expenses on collecting items?

Are partners of collectors aware of the fact that there are differences between expenses declared to them and the real expenses on collecting items?

We suppose that:

**H1.** 
*Women declaring regular expenses on collecting items, in comparison to women who do not declare such expenses, hide amounts of money which they spend on collecting items.*


**H2.** 
*Men declaring regular expenses on collecting items, in comparison to men who do not declare such expenses, hide amounts of money which they spend on collecting items.*


**H3.** 
*There are differences between women and men, declaring regular spending of money on collecting items, in hidden amounts of money spent on collecting items.*


**H4.** 
*There are differences between collectors earning more and less than the national average in the amounts of money regularly spent on collecting items.*


**H5.** 
*There are differences between women and men, declaring regular spending of money on collecting items, in techniques of hiding their expenses on collecting items.*


**H6.** 
*Partners of people who regularly spend money on collecting items are aware of the fact that there are differences between the amounts of money spent on collecting items declared to them and the real ones.*


The sources of such hypotheses are, among others, studies showing that men, unlike women, spend more and tend to risk more money on the stock exchange [23]. What is more, men, when investing, more often hide the real value of their investments and they perceive the money spent differently [24]. I thus suppose that there will be differences in the amounts of money allowed for their own pleasures by men and women. Other studies show that men more often lie in situations bringing them benefits [25]. These results were also confirmed by other studies [26]. Women lie more often because they do not want to harm others. That is why they hide the truth in a more subtle way more often than men [27]. Men try to achieve their goal and a lie can be a means to do it. As far as women are concerned, they hide their lies more often as they are afraid that other people’s opinion about them will become worse. When men lie, they act in a more selfish way than women, and they often cheat [28]. They are more aggressive and independent in their actions, whereas women pay more attention to partnership. They are subject to it [29]. Moreover, some studies [30] show that we detect lies better in situations which are familiar to us. Lies are detected more efficiently by these people who have known the liar for a long time and, thanks to that, have experience in detecting their lies [31]. That is why, despite the fact of hiding real expenses, partners of people spending money on their pleasures (collectors) know that they are being deceived.

## 2. Materials and Methods

When testing the hypotheses, research has been carried out in order to show whether and how men and women (collectors, that is, people who regularly spend money on collecting items) deceive their partners when informing them about the amount of money spent on their own pleasures. Six main statistical analyses have been carried out. The first one aimed at determining whether there are differences among women who declare that they spend money on collecting items and those who do not do that. The second one, similarly to the first one, checked if there are differences among people who declare that they spend money on collecting items and those who do not do that. However, it concerned men. The third one analysed the differences among men and women in lowering the declared expenses on the collecting items. The fourth one verified if there are differences among collectors earning more and less than the national average in the amounts of money regularly spent on their own pleasures. The fifth one aimed at checking whether there are differences in the strategies used by men and women in order to hide their real expenses on collecting items. The final sixth one checked if partners of questionnaire participants know that the amounts of money spent on collections and stated by questionnaire participants are false.

Additionally, three qualitative analyses were performed. Two of them concerned expense concealment techniques. The third concerned the collector’s motivation for lying to his or her partner.

### 2.1. The Questionnaire Participants and the Research

The research sample included 120 people (68 men and 52 women: M age = 35.3, SD = 4.7) who declared having a hobby related to collecting items (or partners of such people) and spending quite regularly (monthly) some amounts of money on their hobby such as model making, collecting, fishing, and shopping (people described this activity as a hobby in the research). Among these people, 37% lived in cities with more than 100,000 inhabitants, and 26% lived in cities with 20,000 to 100,000 inhabitants. The last group of respondents (37%) lived in towns with less than 20,000 inhabitants. About 96% of the respondents have a permanent source of income. About 75% of the respondents earned less than PLN 5000 (EUR 2500) per month, and 25% earned more than PLN 5000 (EUR 2500). In addition, 25% of people had higher education, 52% had secondary education, and 23% had vocational and primary education. Questionnaire participants did not call themselves “collectors” but declared regular spending of money on their pleasures, and this was the criterion for counting them as members of the group of collectors. They are called “collectors” in the research. These people were found in clubs of collectors, reconstruction groups, at gatherings of completists, or at mass events. A control group was formed by people who declared that they do not spend their money systematically on their pleasures. Every research participant was asked to fill in a questionnaire. The second stage was a request to deliver and fill in a questionnaire by the research participant’s partner.

### 2.2. Variables and Their Operationalisation

Collecting is, as Belk [32] writes, “an intensely engaging form of consumption”. It is focused on acquiring and possessing [33]. Statistical data collected by marketing companies [34] in the United States show that 6 out of 10 Americans are collectors, spending an average of about USD 6200 on their collections. In the presented research, the amount of money spent on collecting items was a dependent variable. It was measured by means of an anonymous questionnaire including questions concerning gender, age, whether somebody can call him/herself a collector, approximate monthly income, kind of hobby, whether money is spent quite regularly on collecting items (on a monthly basis) and what kind of amounts of money are spent, whether their partner knows how much money is really spent monthly on that hobby, and if not, what kind of strategies are used by a research participant to hide expenses. Based on the answers received, 18 different strategies for hiding expenses can be identified. Using the terminology of defence mechanisms among addicts [35], the strategies of hiding expenses were divided by competent judges into four conventionally named categories, of which two dominated. In the first one, preventive strategies, a research participant concedes to their partner to buying, but at the same time, lowered the cost of the purchase, e.g., by saying that it was an exchange, a donation, or that he/she was paying for a service or that the price was lower than it really was. In the second one, denying strategies, a research participant does not inform their partner about the purchase, and after the partner finds out about the purchase, the research participant continues to deny it, e.g., by saying that it was purchased a long time ago, that it was borrowed, or that he/she has had it but it was hidden. In addition, the questionnaire included questions on why the research participant hides real expenses and if he/she feels as if he/she was deceiving the other party; whether, according to the research participant, the other party knows that he/she is deceived; and whether the partner also collects items and if he/she reveals the real amount of money spent on them. The questionnaire had 26 questions in total. The procedure of collecting the filled-in questionnaires was made in such a way that in most cases, in order to carry out statistical analyses, it was possible to group people being in the same relationship. Having or not having a hobby (related to collecting items) on which the research participant spends money regularly was an independent variable.

The control group also received a questionnaire identical to that of the research group.

The next stage was the personal delivery of the questionnaire to the collectors’ partners. It contained questions, among others, on whether collectors inform their partners about the amounts spent on collecting and whether, in their opinion, these amounts are underestimated, true, or overstated.

## 3. Results

In order to verify the hypotheses, seven detailed statistical analyses were carried out.

The first one, carried out with the Chi-square model in order to verify H1, showed that there are differences in hiding one’s expenses between women declaring systematic spending of money on their own pleasures (collections) in comparison to women who do not declare such expenses (Chi-square = 7.06; *p* < 0.05). The other one, also carried out with the Chi-square model, showed that there are differences in hiding one’s own expenses between men declaring systematic spending of money on their own pleasures (collections) in comparison to men who do not declare such expenses (Chi-square = 12.08; *p* < 0.05). The third analysis, carried out with the model of t-test (men, women) × 1 (the amount of money spent monthly on collecting items), in order to verify H3, showed that the compared groups significantly differed statistically [F(4.112) = 9.03; *p* < 0.05], and the effect is equal to n^2^ = 52.4%. Men spend more on their collections. A more detailed analysis showed that men declared a lower value of real expenses on their collections on average by 30%. Men declaring that they spend less than PLN100 (EUR 50) monthly on collecting items did not appear in the study. The lowest amounts of money (from PLN 100 (EUR 50) to PLN 1000 (500 EUR) monthly) were the least lowered, that is, by about 15%, whilst the highest amounts (over PLN 1001 (EUR 501) monthly) were the most lowered, that is, by about 38% (see Figure 5). Over PLN 2300 (EUR 1150), men stopped informing their partner about their next expenses. The women who took part in the research lowered their expenses on average by 45%. In 23% of cases, they declared that they lower the amounts they spend on collecting items not by the amount but to the amount of PLN 200 (EUR 100). The fourth analysis, carried out in the model of t-test (earning more than the national average, earning less than the national average) × 1 (what percentage of salary is spent monthly on collecting items), in order to verify H4, showed that the compared groups significantly differed statistically [F(4.112) = 7.32; *p* < 0.05], and the effect is equal to n^2^ = 51.1%. It showed that the obtained monthly income differentiates expenses on one’s item collections.

People who have higher income than the national average (PLN 5000 (EUR 2500)) spend about 40% of their monthly income, whereas people earning less than the national average spend about 28% of their monthly income. The fifth and sixth analyses, in order to verify H5, carried out with the Chi-square model, proved that women use denying techniques more often than men (Chi-square = 21,34; *p* < 0.005), whereas men use preventive techniques more often than women (Chi-square = 19.67; *p* < 0.005). An additional analysis, in order to verify H6 (Chi-square = 18.27; *p* < 0.05), showed that partners are aware of the fact that they are deceived as far as the real amounts of money spent on collecting items are concerned.

The last two analyses were qualitative. The first one showed what kind of techniques are used by collectors in order to hide not only amounts of money but also the value of collected items. The most interesting techniques of deceiving partners which were mentioned in the pilot research are as follows:

“I order a glass case and ask the seller to cover it a little with dust. Then, I tell my wife that I have just taken it from the basement where it spent quite some time.”

“I ask the seller to send me a partial invoice with a lower price and then, I show it to my wife—she is never angry with me about it.”

“When I buy, the seller writes on the parcel the word ‘fish’ and then, the postman knows that he is not supposed to give the parcel to my wife, but to me personally.”

“I order goods to be sent to my brother’s/uncle’s/mother’s”

“I ask to write on the parcel the words ‘transformer at the bridge’. The postman knows that if he sees words like this on the parcel, he is supposed to deliver it to an agreed spot.”

“I go to the seller earlier and pay him PLN 300 (EUR 150). I come to this shop later with my wife and give the seller the remaining amount of money—PLN 100 (EUR 50). In this way, my wife never makes a fuss about it as she thinks that this item cost only PLN 100 (EUR 50).”

An additional qualitative analysis of the questionnaire concerned the motivation of lying to a partner. It showed that collectors do not tell the truth about their expenses because they feel that their partners do not understand their hobbies.

## 4. Discussion

The obtained results showed that collectors (people who regularly spend money on item collections), both women (positive H1 verification) and men (positive H2 verification), hide real expenses on this kind of pleasure. Moreover, there are differences between sexes as far as the amount of money spent on pleasures is concerned (positive H3 verification). Men spend more on item collections, but women are the ones who lower the amounts more (by about 45%) than men (by about 30%). It is possible that their hobbies cost more. There was an interesting case of three men who declared expenses higher than PLN 500 (EUR 250) monthly, but each of them hid their hobby and did not inform their partner about the expenses, which can be considered as an addiction to collecting items. Two of them took an extra job (the third one set up a business) that the partner did not know about and this was caused by expenses connected with the hobby of collecting. All of them collected rare and valuable coins. It is worth considering how thin the line is between a hobby of collecting and an addiction. Another conclusion of the research (positive H4 verification) is that affluence is related to the money spent on collections. The higher the income is, the more money is spent, which seems to be a logical conclusion. However, regardless of earnings, 34% of money earnt is spent on hobbies. It is possible to say that money limits collecting. If there was more money, more of it would be spent on items. However, is there a top boundary after the exceeding of which collectors would say, “stop, I do not spend more”, or does collecting act as a drug, causing the addicted person to want more and more? Subsequent studies could provide us with an answer to this question. The obtained results showed that there are differences between sexes in strategies of hiding expenses (positive H5 verification). Men prefer prevention, that is, informing their partner earlier about a new purchase, but at the same time, they lower the real cost of this purchase, whilst women prefer the denying style, that is, even after their new purchase is revealed, they do not concede to having bought it. Women try not to inform their partners about their expenses as they believe that in the event of discovering them, they will use *denying* techniques (that it was bought a long time ago or it was borrowed, etc.) Men rather try to avoid negative consequences in the event of revealing new expenses by informing about the expenses, but they use preventive techniques (such as an exchange; lowering of the price). It is possible that the causes of such behaviours lie in gender differences, empathy, and speaking in a more detailed way in two dimensions [36]: agency (i.e., concern for self) and communion (i.e., concern for others). According to some studies [37], women pay more attention to surroundings and partnership than men and thus, they avoid conflicts whenever it is possible. Women do not inform about their expenses, as such information could result in conflicts with their partner. They believe that the expense itself will not be noted and thus, they decide to act only after it has been revealed. Men, unlike women [37], are focused on themselves and their possibilities of succeeding. Their *preventive* technique, despite the initial discomfort felt by their partner connected with the information about the expense, nullifies the danger connected with its later revelation and at the same time, contributes to their success. Another explanation can be stereotypical perceptions of women and men. As classic studies show [38], stereotypes can cause a given person to be perceived as incompetent in a given area. A lot of studies on stereotypes have been published to date. Some of them [39] show that there is a stereotype of a woman who, unlike a man, has no mathematical skills, cannot analyse complex problems logically, and does not think too much about the substance of problems, but is able to note details and even subtle changes in the surroundings [40]. It is thus possible that men try to inform about a new purchase as a preventive technique because their partner would notice it anyway, but at the same time, they lower its price, believing that their partner will not analyse the information presented by them in detail. Women act differently. They believe that men are not very observant and that thanks to that, they will avoid the need to inform them about a new purchase. It is worth mentioning a new tendency between sexes which requires further exploration. Men lower the price of the product they bought, but after exceeding PLN 2300 (EUR 1150), they do not inform their partner about another purchase. Women also use another mechanism besides lowering the price. They tell their partner that a single purchase did not exceed a given amount (the most frequent amount is PLN 200(EUR 100)). This suggests that men, while describing their expenses, think while taking into consideration the time frame, that is, they try not to exceed a given amount of money set by themselves, e.g., the amount of monthly expenses. Women, however, use the technique of lowering the price and also the technique of informing that their single expense did not exceed a given amount. However, this interesting result requires further research.

A conclusion of the research which gives us food for thought (positive H6 verification) is the fact that despite the efforts of collectors, their partners know that they are deceived, despite the fact that they use preventive methods.

The final analyses were qualitative. They showed how creative collectors are when they hide their expenses.

There is one more important issue that should be described in more detail in future research. It is the motivation to cheat on one’s partner. Lying comes at a cost to one’s self-esteem or morality, and is often rationalised to maintain a positive self-image [41]. The sender is often not the only beneficiary of the lie, and the receiver is often not the only person harmed [42]. As some studies show, people often lie about their feelings [43] and can use them as a rationalisation for their actions. A similar mechanism is evident in these studies. According to the respondents’ declarations, the motivation to cheat stems from his/her lack of understanding of the partner’s hobby. These are the preliminary results of the research, which will be enriched with further analyses and publications in the near future.

## 5. Conclusions

Collectors of all genders do not tell the truth about their collecting expenses to their partners. They also lower the declared amounts for collecting. Men and women differ in the techniques of hiding their expenses, but they hide them in an equally creative way.

This creative approach of people who are deceiving their partner shows that the pleasure of spending money on item collections is stronger than honesty towards their partner. It is possible to ask here about the motivation to act like this. An answer to this question can be found also thanks to a qualitative analysis. According to declarations of the research participants, the reason for deceiving their partner is the partner’s lack of understanding of their hobby of collecting. It is thus possible that the acceptance, or at least tolerance, of the partner’s passion could lead at least to decreasing the deceiving effect, if not its complete disappearance. Notably, people who are deceived are aware of the fact that they are deceived. It is a kind of double play. One person lies to the other with his/her eyes open, and the other person knows that he/she is deceived and allows the partner to continue doing it. These are the initial results of the research and they will soon be enriched with other analyses. It is also necessary to mention the limitations of the conducted research. Only declarations were examined, not the actual behaviour of collectors. It should also be remembered that the questions contained in the questionnaire concerned sensitive issues, such as, for example, cheating on a partner, so the respondents could lie when answering such questions. Unfortunately, it was not possible to reach a larger group of collectors; therefore, only a small group of them was examined, which does not necessarily give full insight into the collectors’ conduct.

To summarise, the research presented here attempts to show a phenomenon of collecting from the psychical point of view, both of a collector and of his/her partner. The results are only an initial elaboration of collected data and not all of it. Pieces of information which are still to be analysed concern, among others, differences in the operation of collectors in case of a common and separate budget in a relationship, seeing one’s own hobby and its influence on the people around them, rivalry among partners having the same hobby of collecting, and moral dilemmas of the deceiver.

## Figures and Tables

**Figure 1 ijerph-19-16755-f001:**
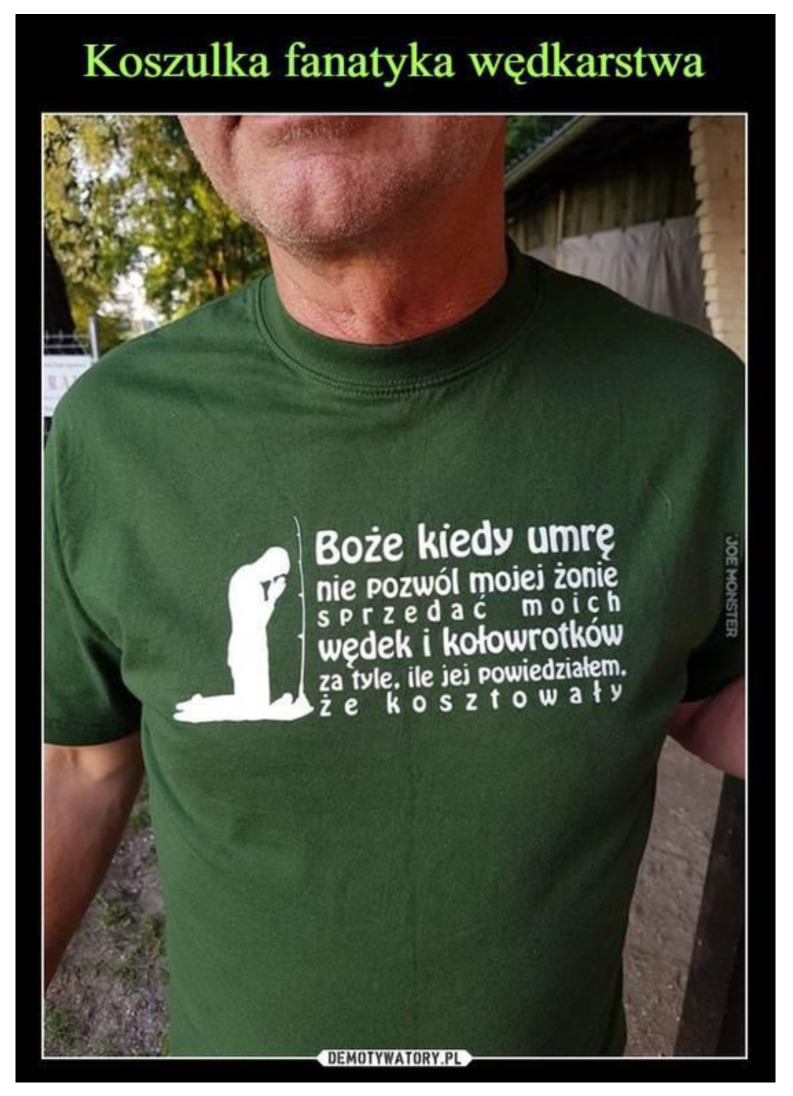
T-shirt of a fishing fanatic.

**Figure 2 ijerph-19-16755-f002:**
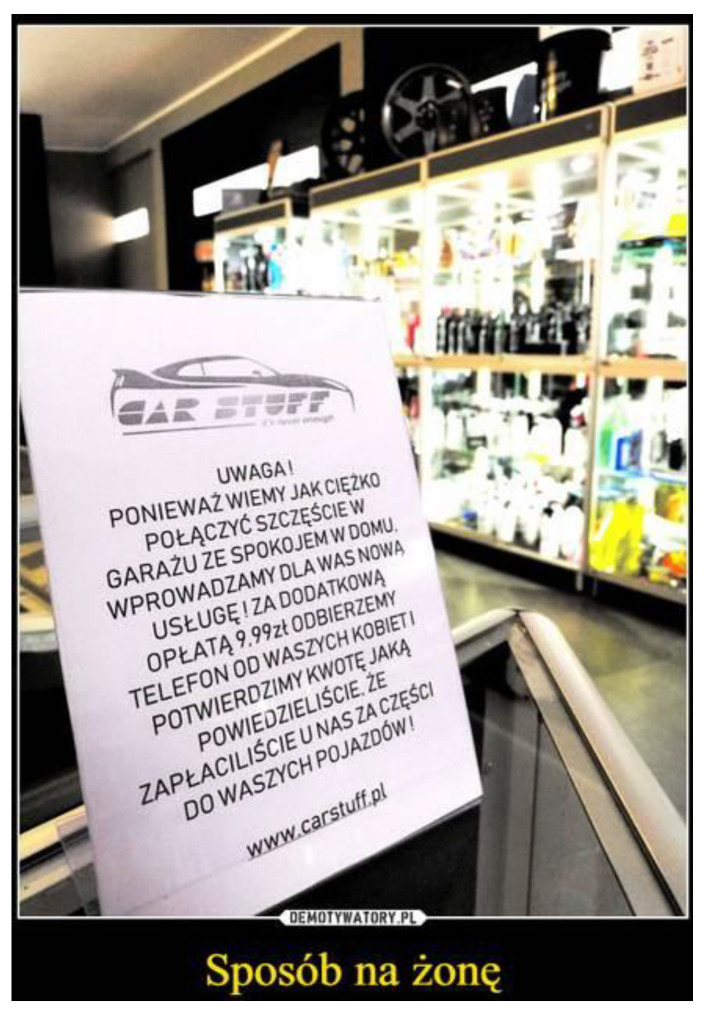
How to deceive your wife.

**Figure 3 ijerph-19-16755-f003:**
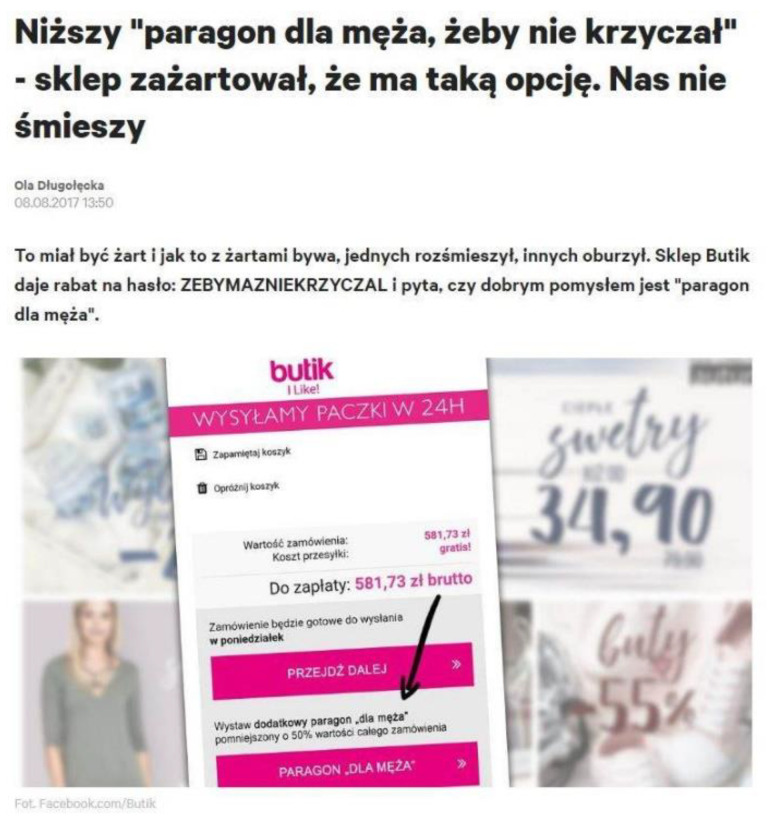
Lower “receipt for a husband so that he did not shout”—a shop made a joke that it has such an option.

**Figure 4 ijerph-19-16755-f004:**
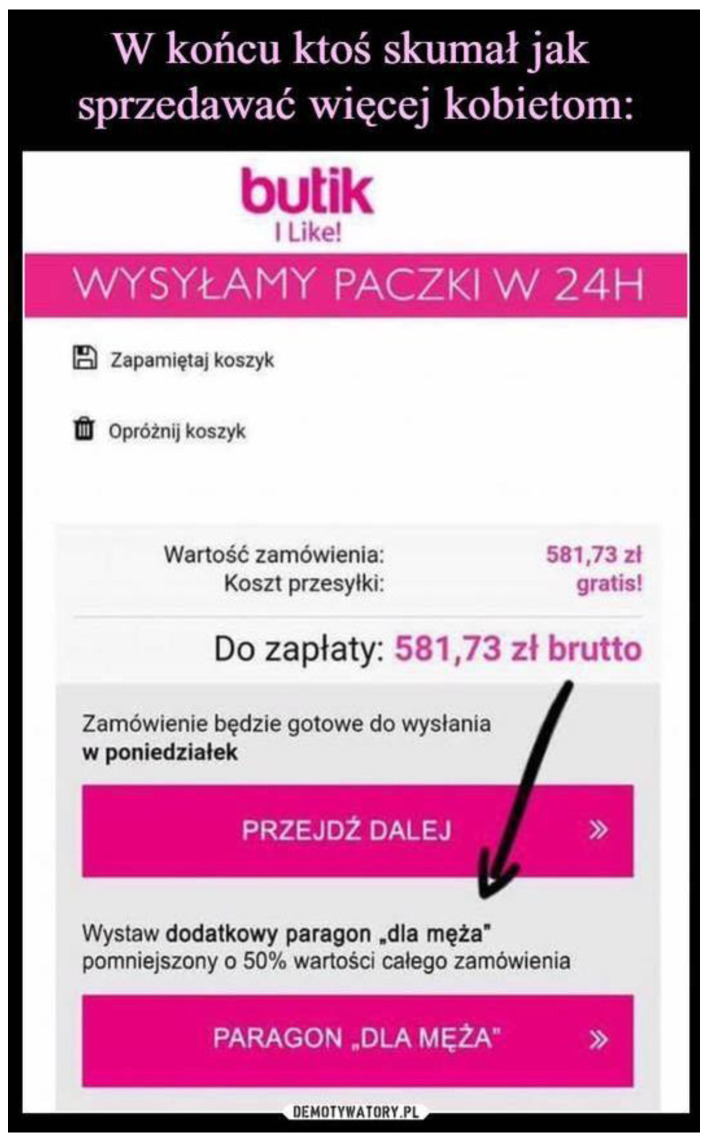
At last, somebody thought of a way to sell something more to women.

**Figure 5 ijerph-19-16755-f005:**
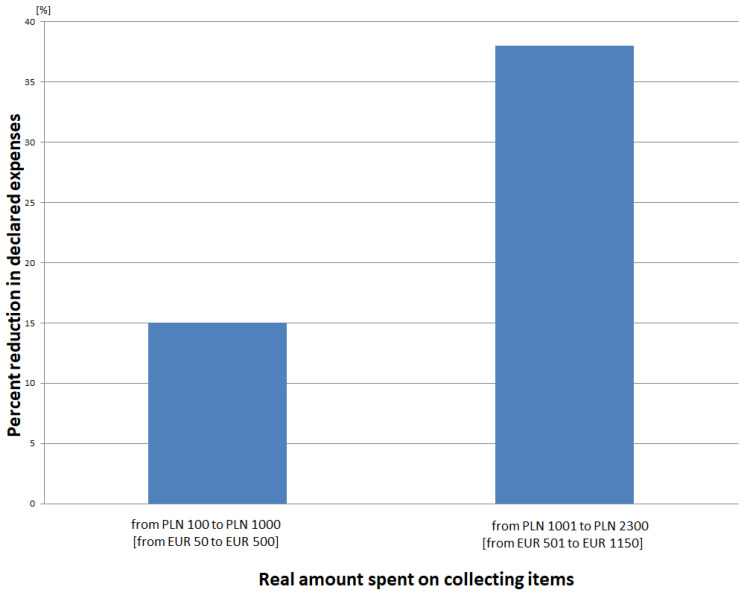
Percentage of reduction in the real amount of money that men spend on collecting items.

## Data Availability

The data presented in this study are available on request from the corresponding author. The data are not publicly available due to privacy restrictions.

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
