# Peer review of "A Collector Deceives—About the Ways of Deceiving Women by Men and Men by Women as far as Spending Money on Collecting Items Is Concerned"

_ijerph, 2022, doi:10.3390/ijerph192416755_

Round 1

Reviewer 1 Report

The paper is generally interesting, and the phenomenon of correlation and discrepancy between collecting and lying that the authors uncover is worth exploring.

However, I think there should be some improvement before this paper is suitable for publication.

First of all, the paragraphs of this paper should be distinguished. The first paragraph of the paper takes up more than two pages, which makes it very hard for the reader to read

Second, before describing the relationship between collecting and lying, it is necessary for the authors to show that collecting is an addictive behavior and the connection between this addictive nature and lying, otherwise it will not fit the main scope of this journal.

Third, the results should be shown with figures or tables. Relying soley on the textual descriptions to present the results makes it difficult for the reader to clearly understand whether the hypotheses are supported by sufficient evidence.

Finally, with regard to the act of lying, moral disengagement is a very important mechanism. Whether collectors have rationalized their lying behavior is worth discussing, and the authors are advised to refer some articles such as (DePaulo, et al., 1996; Xiong et al., 2022)

DePaulo, B. M., Kashy, D. A., Kirkendol, S. E., Wyer, M. M., & Epstein, J. A. (1996). Lying in everyday life. Journal of Personality and Social Psychology, 70(5), 979–995.

Xiong, X., Zhang, Y., & Zhou, X. (2022). Adding a third-party player in the sender-receiver deception game. Current Psychology, 1-14.

Author Response

Response to Reviewer 1 Comments

Point 1: First of all, the paragraphs of this paper should be distinguished. The first paragraph of the paper takes up more than two pages, which makes it very hard for the reader to read

Response 1: The introduction has been divided into 12 paragraphs which makes the chapter more clear

Point 2 Second, before describing the relationship between collecting and lying, it is necessary for the authors to show that collecting is an addictive behavior and the connection between this addictive nature and lying, otherwise it will not fit the main scope of this journal.

Response 2: The relation between collecting and addiction has been taken into account and supported with the literature

Point 3

Third, the results should be shown with figures or tables. Relying soley on the textual descriptions to present the results makes it difficult for the reader to clearly understand whether the hypotheses are supported by sufficient evidence.

Response 3: Data has been visualised on the graph

Point 4

Finally, with regard to the act of lying, moral disengagement is a very important mechanism. Whether collectors have rationalized their lying behavior is worth discussing, and the authors are advised to refer some articles such as (DePaulo, et al., 1996; Xiong et al., 2022)

DePaulo, B. M., Kashy, D. A., Kirkendol, S. E., Wyer, M. M., & Epstein, J. A. (1996). Lying in everyday life. Journal of Personality and Social Psychology, 70(5), 979–995.

Xiong, X., Zhang, Y., & Zhou, X. (2022). Adding a third-party player in the sender-receiver deception game. Current Psychology, 1-14.

Response 4: Comments on the motivation for cheating on a partner have been included in the text (based on the proposed and extended literature)

Reviewer 2 Report

It is an interesting topic, and the authors presented a very interesting introduction. It can be found that the authors made some mistakes in the statistics results presentation section.

Line 263, Ch2 model should be Chi-square test.

Line 266, Chi2 should be Chi2 or Chi-square. As well as n2 should be n2.

Does PLN 100/500 refer to the amounts from PLN100 to PLN500?

The authors declare that the respondent's information should be kept private. But the respondents' demographic information is of great importance for readers to judge if the results are reliable. I recommend that authors should supplement the basic information of the respondents, such as: income, age, education experience or other related information.

Author Response

Response to Reviewer 2 Comments

Point 1: Line 263, Ch2 model should be Chi-square test.

Response 1: this correction has been applied to the text

Point 2 Line 266, Chi2 should be Chi2 or Chi-square. As well as n2 should be n2.

Response 2: this correction has been applied to the text

Point 3

Does PLN 100/500 refer to the amounts from PLN100 to PLN500?

Response 3: Yes, this correction has been applied to the text

Point 4

The authors declare that the respondent's information should be kept private. But the respondents' demographic information is of great importance for readers to judge if the results are reliable. I recommend that authors should supplement the basic information of the respondents, such as: income, age, education experience or other related information.

Response 4: the article was supplemented with demographic data

Reviewer 3 Report

I believe that the area of research will trigger interest among the readers. The highlighted issue is very rare in the context of consumer behaviours. However, some modifications is needed to suit with the scientific content of a journal manuscript. I therefore suggest you to improve the following:

1.     Section 1 – The discovery of collectors behaviours and lies was interesting, however, please put in few separate paragraph and to restructure it accordingly (synchronisation).

2.     Please check the term “I” (line 157) vs the term “we” (line 159/169). 

3.     I have detected these terms in the manuscript (for example in line 155 to line 221: collectors/people who regularly spend money on their own pleasure; pleasures/hobbies; pleasures (hobbies); pleasures; pleasures (collectors), which will lead to confusion among the reader. Does pleasures mean the same to hobby? What about collectors? I hope you can revise the term by properly define it, and to standardise the usage of the term in the manuscript. 

4.     Section 2.1 and Section 2.2 – I think you need to properly explain the steps taken for the method because it is quite confusing. Survey vs pilot research vs questionnaire vs second stage – which one is the first stage? You may want to separate the pilot study and the real data collection explanation to avoid confusion. How did you deliver the questionnaire (line 233)? What does statement on line 234-235 means?

5.     Section 2.2 – How did you develop the variables? What is the underpinning theory for the variables? I think more recent literature is needed for the whole research.

6.     Line 261 – seven detailed statistical analyses is contradicted with statement on line 209.

7.     Please check the term use for statistical analysis in Section 3 whether it is the common term use worldwide – Ch2 model and t-student.

8.     Do you want to consider to convert PLN to either Euro or USD for international reader? Perhaps it will attract more citation.

9.     I would like to suggest you to add one section for discussion, because right now you are combining your discussion in the conclusions section.  

10. Please also support your discussion with recent literature and to break your discussion into few paragraphs.

11. You have mentioned qualitative on line 357 but it was not mention in your method section. Please revise. Again, you use pilot research on line 360, hence, it’s very confusing. 

12. You should put the transcribe (line 361 to 373) on result section. 

13. Please make sure that your conclusion relates with the issue and aim of your research.

14. Please also consider to add limitation of your research and how does it contribute to related parties. 

15. Please revise the abstract according to a common scientific content.

Author Response

Response to Reviewer 3 Comments

Point 1: Section 1 – The discovery of collectors behaviours and lies was interesting, however, please put in few separate paragraph and to restructure it accordingly (synchronisation).

Response 1: The introduction has been divided into 12 paragraphs which makes the chapter more clear

Point 2 Please check the term “I” (line 157) vs the term “we” (line 159/169).

Response 2: this correction has been applied to the text (we used “we” version)

Point 3

I have detected these terms in the manuscript (for example in line 155 to line 221: collectors/people who regularly spend money on their own pleasure; pleasures/hobbies; pleasures (hobbies); pleasures; pleasures (collectors), which will lead to confusion among the reader. Does pleasures mean the same to hobby? What about collectors? I hope you can revise the term by properly define it, and to standardise the usage of the term in the manuscript.

Response 3: the article submitted for review concerns people collecting items so the terms "hobby" and "pleasure" have beed converted into “collecting items ".

Point 4

Section 2.1 and Section 2.2 – I think you need to properly explain the steps taken for the method because it is quite confusing. Survey vs pilot research vs questionnaire vs second stage – which one is the first stage? You may want to separate the pilot study and the real data collection explanation to avoid confusion. How did you deliver the questionnaire (line 233)? What does statement on line 234-235 means?

Response 4: There were errors in terminology during the translation stage. The terminology has been unified. The questionnaires were delivered in person.

Point 5

Section 2.2 – How did you develop the variables? What is the underpinning theory for the variables? I think more recent literature is needed for the whole research.

Response 5: The operationalization of variables has been improved, enriched with theories and newer literature.

Point 6

Line 261 – seven detailed statistical analyses is contradicted with statement on line 209.

Response 6: The inaccuracy has been corrected

Point 7

Please check the term use for statistical analysis in Section 3 whether it is the common term use worldwide – Ch2 model and t-student.

Response 7: The terms has been corrected

Point 8

Do you want to consider to convert PLN to either Euro or USD for international reader? Perhaps it will attract more citation.

Response 8: PLN has been converted into EUR.

Point 9

I would like to suggest you to add one section for discussion, because right now you are combining your discussion in the conclusions section.

Response 9: The article now includes a discussion and conclusion section

Point 10

Please also support your discussion with recent literature and to break your discussion into few paragraphs.

Response 10: The disscussion section has been updated with newer literature.

Point 11

You have mentioned qualitative on line 357 but it was not mention in your method section. Please revise. Again, you use pilot research on line 360, hence, it’s very confusing.

Response 11: The text has been corrected as suggested

Point 12

You should put the transcribe (line 361 to 373) on result section.

Response 12: The transcript has been moved to the result section.

Point 13

Please make sure that your conclusion relates with the issue and aim of your research.

Response 13: After enriching the data with the motivation of cheating partners, the conclusion should already correspond to the aim of research

Point 14

Please also consider to add limitation of your research and how does it contribute to related parties.

Response 14: The limitation of research details have been added

Point 15

Please revise the abstract according to a common scientific content.

Response 15: Abstract has been improved

Round 2

Reviewer 1 Report

I would like to acknowledge the authors' efforts in the revision of this manuscript and am glad to see that the manuscript has been improved a lot.

Thanks for the revision and good luck with your research.